# Smart Dental Materials Intelligently Responding to Oral pH to Combat Caries: A Literature Review

**DOI:** 10.3390/polym15122611

**Published:** 2023-06-08

**Authors:** Kan Yu, Qinrou Zhang, Zixiang Dai, Minjia Zhu, Le Xiao, Zeqing Zhao, Yuxing Bai, Ke Zhang

**Affiliations:** 1Department of Orthodontics, Beijing Stomatological Hospital, School of Stomatology, Capital Medical University, Beijing 100050, China; 2School of Stomatology, Chongqing Medical University, Chongqing 401147, China; 3Department of Stomatology, Beijing Friendship Hospital, Capital Medical University, Beijing 100050, China

**Keywords:** smart materials, dental resin, pH-responsive, antibacterial, biofilm, drug delivery system, caries

## Abstract

Smart dental materials are designed to intelligently respond to physiological changes and local environmental stimuli to protect the teeth and promote oral health. Dental plaque, or biofilms, can substantially reduce the local pH, causing demineralization that can then progress to tooth caries. Progress has been made recently in developing smart dental materials that possess antibacterial and remineralizing capabilities in response to local oral pH in order to suppress caries, promote mineralization, and protect tooth structures. This article reviews cutting-edge research on smart dental materials, their novel microstructural and chemical designs, physical and biological properties, antibiofilm and remineralizing capabilities, and mechanisms of being smart to respond to pH. In addition, this article discusses exciting and new developments, methods to further improve the smart materials, and potential clinical applications.

## 1. Introduction

Dental caries is the most prevalent disease in humans. Dental plaque secretes acids to cause caries, leading to the destruction of dental hard tissues, resulting in significant economic and health burdens worldwide [1]. The development of dental caries is a comprehensive and dynamic process involving the interaction of food, microorganisms, and host factors, including the equilibrium of dental hard tissue demineralization and remineralization. Changes in local pH caused by acid-producing bacteria such as Streptococcus mutans (*S. mutans*) play a key role in the caries process [2,3]. Caries initiates with the dissolution of hydroxyapatite (Ca_10_(PO_4_)_6_(OH)_2_), the primarily component of enamel and dentin. Bacteria such as *S. mutans* and Lactobacillus produce acids through metabolism, resulting in a low pH environment on tooth surfaces. The teeth then demineralize with the loss of Ca^2+^ and PO_4_^3−^ and the destruction of tooth structure.

Efforts have been made to improve tooth cavity filling materials, including glass ionomer cements (GIC), resin-modified glass ionomer cements (RMGIC), and composites. However, secondary caries is a main problem after treatments, due to the incomplete removal of infected dental tissue, microleakage, and poor oral hygiene habits. Although materials such as GIC can release fluoride ions, there has been no significant improvement in preventing secondary caries. Developed two decades ago, Ariston pHc (Ivoclar Vivadent) can release more ions at low pH with the ability of release-recharge and re-release [4]. More recently, a new generation of smart materials has been developed that can neutralize acids, release therapeutic ions, suppress biofilm growth, minimize biofilm acid production, regulate the oral microbial community to prevent caries, protect enamel hardness, trigger antibacterial activity only at low pH, and treat periodontal inflammation based on enzyme response systems [5,6,7]. Smart dental materials are known as such because they can function in response to specific changes occurring in the oral environment, such as light, pH, and temperature. A previous review article focused on smart biomaterials and constructs for hard tissue engineering and regeneration, especially on bone regeneration [8]. The present review focuses on smart dental materials that respond to pH to combat caries, which is different from the previous review article [6,7,9,10,11] (Figure 1).

We searched papers published between 2015 and 2023 based on keywords such as caries, dental resins, smart materials, and antibacterial, and screened them according to their relevance to the topic. Montoya et al. introduced some smart dental materials for antimicrobial applications; others have also introduced smart materials generally, but no one has concentrated on smart dental materials that respond to pH to combat caries [5,9,10,11]. A literature search revealed no review papers to date on smart pH-responsive dental resin materials. The present study represents the first review article on the development and mechanisms of smart pH-responsive dental resin materials. This article reviews the recent cutting-edge research on smart pH-responsive dental resin materials and discusses their novel microstructural and chemical composition designs. Furthermore, this article reviews their physical and biological properties, as well as antibiofilm and remineralizing capabilities. In addition, this article addresses the underlying mechanisms of being smart to respond to local oral pH, and the potential clinical applications to combat caries and promote oral health.

## 2. Resins That Can Inhibit Bacterial Acid Production and Raise the pH

Most oral bacteria grow best in a pH-neutral environment. A decrease in pH inhibits the normal flora, while some acid-producing and acid-tolerant bacteria become the dominant species. For example, *S. mutans* survive at pH below 5.0 and could degrade various monosaccharides and polysaccharides to produce acid, reducing the pH further [12]. A rapid decline in plaque pH and a sustained low pH environment often cause the demineralization of dental hard tissue [13]. Therefore, it is important to maintain the stability of oral local pH and inhibit the growth of acid-producing and acid-tolerant bacteria.

Nanoparticles of amorphous calcium phosphate (NACP) is the nano-morphology of amorphous calcium phosphate and is also the precursor of apatite [14]. The average particle diameter of NACP is 116nm (Figure 2a), and its specific surface area is 35 times that of ACP [15]. Formed by adding NACP, 5% 2-methacryloyloxyethyl phosphorylcholine (MPC), and 5% dimethylaminohexadecyl methacrylate (DMAHDM) to PEHB resin, a novel dental material can simultaneously have triple advantages of calcium and phosphate ions recharge, as well as protein-repellent and antibacterial capabilities [16]. NACP can release supersaturated levels of Ca^2+^ and PO_4_^3−^, and the lower the pH, the more ions are released, especially at the cariogenic pH of 4 (Figure 2b) [17]. The released Ca^2+^ and PO_4_^3−^ can form hydroxyapatite, which is a putative mineral in enamel and dentin [18]. The Ca-PO_4_ composite materials have a great efficacy for remineralizing enamel and dentin lesions in vitro [19]. Furthermore, NACP has a strong ability to neutralize acids and increase pH, which can restore the solution pH from 4 to more than 5.5 (Figure 2c) [20]. Co-culture with oral saliva-derived dental plaque biofilm for 72 h can increase the pH to a safe level of 6 or even more, and the higher the NACP filler mass fraction, the stronger the acid neutralization ability (Figure 2d) [21]. The mechanism of its acid neutralization effect is as follows: the living bacteria in the plaque first produce acid to reduce the pH of the medium. In this process, NACP will neutralize the acid at the same time and slow down the speed of pH decrease. After 8 h of co-culture, the sucrose supplement in the biofilm medium may be exhausted, and the acid production capacity of the bacteria is weakened. NACP then continues to neutralize the remaining acid in the medium and increase the pH [16]. Meanwhile, NACP could inhibit the growth of *S. mutans* (Figure 2e). Pyromellitic glycerol dimethacrylate (PMGDM), which is an acidic monomer with carboxylic acid groups, is the main part of PHEB resin and consists of 44.5% PMGDM, 39.5% ethoxylated bisphenol A dimethacrylate, 10% hydroxyethyl methacrylate (HEMA), and 5% bisphenol A glycidyl dimethacrylate. PMGDM from the PEHB resin could chelate with calcium ions from the exterior environment, so as to realize the calcium and phosphate ions recharge-release function [21]. In addition, some components with a nucleation template effect in collaboration with NACP can significantly improve the remineralization capability of NACP [22].

Furthermore, MPC and DMAHDM may cooperate with NACP to resist bacteria. MPC in resin can reduce protein adsorption and bacterial adhesion, slow down the formation of oral acquired pellicle, inhibit the formation of dental plaque biofilm, and tune the ion release by altering the amount of MPC [23,24,25]. DMAHDM is one of the quaternary ammonium methacrylates (QAMs), whose antibacterial effect is mainly known for contact-inhibition. When negatively charged bacteria contact the positively charged sites of QAMs, the electric balance of the cell membrane could be disturbed, resulting in bacterial death [26,27]. In short, PEHB-NACP-MPC-DMAHDM resin can not only increase the pH to a safe zone, but also greatly reduce protein adsorption, biofilm metabolic activity, and plaque growth [16].

Ion-releasing materials such as GIC and RMGIC often show a decrease in mechanical properties over time. Therefore, the new materials use nanotechnology with fine nanoparticles to minimize any flaw size and void size, as well as stable glass particles for mechanical reinforcement. For example, Moreau et al. showed that nanocomposites with NACP and glass reinforcement had a mechanical strength greater than commercial materials already used in the clinic, even after two years of water aging [28]. Meanwhile, the wear resistance of NACP nanocomposite was similar to commercial control materials. Weir et al. developed nCaF_2_ nanocomposites with mechanical properties far exceeding commercial control materials, while having wear resistance matching commercial control materials [29]. In conclusion, it is a valid concern that the exchange of ionic components could imply a lower structural stability of the material. This challenge could be overcome via nanotechnology and by using reinforcement fillers in the material.

In addition to NACP, tetracalcium phosphate (TTCP: Ca_4_(PO_4_)_2_O) also releases Ca^2+^ and PO_4_^3−^, and the release amount at pH 4 is significantly greater than that at pH 7.4 [30]. L-arginine added to the resin can also be released and metabolized by certain bacteria to generates ammonia, which can increase plaque pH and inhibit acid-producing cariogenic bacteria [30,31]. Other bioactive glass can release Ca^2+^ and PO_4_^3−^ as well and increases the pH (more detailed explanation in Point 5).

If the acidic environment is not controlled, the plaque will be repeatedly acidified. The continuous decline of pH may lead to the inhibition of the growth of normal flora, and acid-producing and acid-tolerant bacteria will become the dominant species, which can survive preferentially and further lead to caries. By releasing Ca^2+^ and PO_4_^3−^, the local pH in the oral environment can be restored, which may help the survival of normal flora and prevent the dominant growth of acid-producing and acid-tolerant cariogenic bacteria through competition or antagonism.

## 3. Suppressing Biofilm Acids and Providing Ions to Increase Enamel Hardness

The acid produced by plaque biofilm metabolizing sucrose leads to a local low pH environment and is the direct cause of tooth surface demineralization [1]. Suppressing biofilm acids, meanwhile, providing ions to increase enamel hardness, may not only reduce or stop the mineral loss of the tooth surface, but also restore the hardness of tooth enamel. A novel resin with antibacterial and remineralizing abilities developed by Zhou et al. was used to produce a new bioactive composite containing DMAHDM and NACP for the first time. DMAHDM inhibits biofilm formation and acid production, while NACP releases Ca^2+^ and PO_4_^3−^ under suitable conditions to promote remineralization (Figure 3) [32]. 3% DMAHDM and 30% NACP were added to the composite, which maintained similar mechanical properties as the commercial composites. When the pH decreased from 7 to 4, the release of Ca^2+^ and PO_4_^3−^ ions increased significantly as a response to the decline in pH. In vitro models of secondary caries showed the enamel hardness of the experimental group containing 3% DMAHDM was 25% higher than that of the control. The experimental group containing 3% DMAHDM + 30% NACP had the best protective effect on the edge of the restoration and maintained the same hardness as that of the healthy enamel group after 21 days. In the in vitro model of root caries recurrence based on *Lactobacillus* and *Candida albicans*, the 3% DMAHDM + 30% NACP group significantly reduced the yield of lactic acid and had the highest dentine hardness compared with the control group [33]. An in vitro enamel demineralization experiment based on *S. mutans* biofilm and 3% DMAHDM + 30% NACP composites also showed the optimal hardness of enamel [34].

Another novel composite material with low shrinkage stress and antibacterial and remineralization properties, composed of urethane dimethacrylate and triethylene glycol divinylbenzyl ether, was developed by Bhadila et al., and it significantly decreased the biofilm colony-forming unit by 4 log orders and the production of lactic acid [35]. In the in vitro demineralization tests of enamel and dentin, the 3% DMAHDM + 20% NACP composite material showed good antibacterial and remineralization effects, and both enamel and dentin maintained maximum hardness through the experiment [36,37].

The combination of DMAHDM and NACP showed great potential in inhibiting the production of acid from plaque biofilms and providing ions to promote remineralization. There has been more evidence provided by several other studies that have demonstrated the wide applicability in other biomaterials. Novel bioactive crown cement and magnetic nanoparticle-containing adhesive show similar antibacterial and remineralizing capabilities [38,39].

## 4. pH-Responsive Antibacterial Resins

Tertiary amine (TA) is an organic base that can be protonated to form QAMs easily because of its strong electron donor property at low pH. TA has been used as a carrier for drug delivery systems based on its pH responsiveness [40]. Farnesol can be released from the drug delivery system co-synthesized with TA via pH activation to inhibit microcosm growth and prevent dental caries [41]. Liang et al. attempted to explore TA as a dental modification material for a long-term antibacterial effect [42]. Two novel TAs (dodecylmethylaminoethyl methacrylate (DMAEM) and hexadecylmethylaminoethyl methacrylate (HMAEM)) were added to the dental resin by the reaction of acrylate groups with the methacrylate groups [42]. The new smart dental resin could kill bacteria in a non-contact manner as TA was covalently grafted onto the resin via the grafting reaction of the group. The novel TA-modified resin adhesives (TA@RAs) acted as an antibacterial agent in a pH-sensitive manner. TA@RAs also showed a great long-term antibacterial stability after repeated protonation–deprotonation processes. When the pH value was lower than the critical value, TA@RAs could rapidly inhibit the production of biofilms and reduce tooth demineralization through protonation, so as to prevent secondary caries. When the pH value returned above the critical value, TA@RAs had little influence on normal flora, which helped maintain the oral microbial diversity (Figure 4) [43]. The two TA possessed no significant difference in their antibacterial effect. However, DMAEM modified resin was more sensitive to pH than HMAEM modified resin. One possible explanation for this is that the protonation of DMAEM with higher pKa is stronger than that of HMAEM at the same pH. In the in vitro experiments, DMAEM modified resin significantly inhibited the growth and acid production of *S. mutans* and *Candida albicans* biofilms, significantly reduced the mineral loss and the depth of lesions in hard tooth tissue, and downregulated the expressions of caries genes, virulence related genes, and pH regulating genes of both species biofilms, further confirming the potential in preventing secondary caries [43].

Because of its excellent performance of pH responsiveness, TA has been used to develop more smart dental materials. A novel pH-responsive sealant was developed by adding DMAEM into the resin-based sealant for the first time, which significantly reduced the acid production of the biofilm and may effectively improve the success rate of fossae sealing by combating microleakage without affecting the mechanical properties [44]. DMAEM was also used to develop new resin infiltrates to treat white spot lesions. Huang et al. developed a new smart resin infiltrate containing DMAEM, which protected tooth enamel against acid attack, making it significantly harder than the commercial control group [45].

TA@RAs has shown a strong and long-term ability to inhibit bacteria. Because resins containing NACP have been proven to have the ability to release, recharge, and re-release ions, the use of NACP in collaboration with TA may be a promising way of preventing secondary caries [46].

## 5. Smart pH-Responsive Dental Resins with Bioactive Glass/Bioglass

Bioactive glasses (bioglass, BGs) have been synthesized as a kind of degradable inorganic non-metallic material (Figure 5a). Different from conventional glass fiber, they can degrade into body fluid and release active ions (Ca^2+^, Si^4+^, PO_4_^3−^, etc.) at the same time. The release of Ca^2+^ and PO_4_^3−^ can help generate apatite (such as hydroxyapatite and fluorapatite), which was able to form a close bond with the host bone to repair bone defects and promote bone regeneration [47]. In oral environments, the apatite can promote the remineralization of the dental hard tissue and repair the demineralized enamel and dentin, which suggests that BGs have high bioactivity. Furthermore, BGs can release Na^+^ and K^+^, which exchange with H^+^ in the solution so that the pH would be raised by the remaining OH^-^. BGs have a great potential in hard tissue regeneration [47].

The mineralization and alkalinization ability of BGs could buffer the fluctuation of the pH and prevent dental caries. When BGs were added into dental resin and immersed in PBS, the pH value of the solution rose abruptly. Especially when the resin was infiltrated with up to 20% BGs particles, the pH value could reach 10.8 and be maintained for 21 days (Figure 5b). Moreover, BGs could form calcium phosphate precipitates on their surface (Figure 5c) [48]. In addition, the composition of BGs could be modified to achieve specific biological effects, such as the release of particular ions. The pH-responsive Zn^2+^-releasing glass particles could release more Zn^2+^ when pH decreases (Figure 5d) and inhibit the growth of *S. mutans* in a slightly acidic environment (Figure 5e), so that it could be used to prevent root caries with low acid resistance [49].

There are three types of BGs, including silicate-based glass, phosphate-based glass, and borate-based glass, according to the different compositions of BGs [50]. Meanwhile, different particles such as CaF_2_, ZnO, and SrO can also be added to BGs, and different effects can be obtained with different components. Low-sodium fluoride-containing bioactive glass (low-NaF-containing BG) contains 12% CaF_2_ and could release F^-^ with good remineralization capability for dental hard tissues. When 45S5 and low-NaF-containing BGs were added in dental resin, respectively [51], it can be seen that both BGs had acid neutralization effect at the beginning by raising the pH value of the lactic acid solution from 4 to 6.0–8.5. However, within 16 days, only the resin with 20% 45S5 remained neutral, and the acid neutralization capacity of the other materials was depleted. Furthermore, both BGs could form apatite precipitates in lactic acid solution and artificial saliva, which was conducive to preventing demineralization of dental hard tissue, promoting remineralization, and sealing the irregular and discontinuous margins. Zinc–calcium–fluoride bioglass is another type of BGs that could gradually release Zn^2+^, Ca^2+^, and F^−^, thereby reducing enzyme activity and inhibiting *S. mutans* [52]. Meanwhile, the zinc–calcium–fluoride bioglass enhanced the release of Zn^2+^ and F^−^ in the acidic environment [52].

BGs have their own advantages in smart pH responsiveness and also exhibit osteogenic, remineralizing, and antibacterial properties. In addition to changing the composition of BGs to achieve different effects, BGs themselves can also serve as carriers to deliver other substances. They are known as porous bioactive glass micro/nanospheres (PBGs) (Figure 5f). The chlorhexidine (CHX)-load PBGs increased the release of CHX when the pH value of the environment decreased, thus realizing pH responsiveness [53]. The silver-indole-3 acetic acid hydrazide (IAAH-Ag)-containing PBGs released 2.8 times Ag^+^ at pH 5.0 and pH 1.2 than at pH 7.4 [54].

## 6. Smart Local Drug Delivery System That Can Respond to pH

Drug delivery systems have been developed as an innovative tool to achieve a sustained, controlled, and localized release of drugs. Traditional drug delivery systems focus on zero-order release and can carry nanoparticles, polymers, peptides, and proteins. Carriers such as resins, hydrogels, and micelles can protect the encapsulated drugs from degradation, so that the drug release can be controlled and protected from the changing environment [55]. Recently, several novel smart drug delivery systems were developed that could interact with environmental stimuli to combat caries.

Several drug delivery systems were used to encapsulate conventional drugs such as CHX and farnesal to enable drugs with intelligent pH-responsive capabilities [8,56]. A CHX-loaded nanomaterial (p(DH)@CHX) with hydrogel carrier 2-(dimethylamino)ethyl methacrylate (DMAEMA)-co-2-HEMA could release CHX in response to the pH (Figure 6a). It showed strong antibacterial effects on S. mutans biofilms (Figure 6b) but no effect on healthy saliva-derived biofilm [57]. As a pH-responsive biomaterial, it is capable of combating cariogenic bacteria species and preventing dental caries. Another drug delivery system loaded with CHX demonstrated similar capabilities [58]. In addition to CHX, farnesal is also an antimicrobial agent of concern, and several farnesal-loaded drug delivery systems have been developed to combat dental caries by intelligently responding to the pH [41,59,60]. Moreover, some pH-sensitive chitosan nanoparticles were developed to load antibacterial drugs [61,62]. These chitosan nanoparticles can prevent drug degradation at physiological saliva pH and release antibacterial agents to combat dental caries immediately when plaque biofilms start producing acid.

Some drug delivery systems could be loaded with several drugs at a time to achieve multiple effects against caries [64]. The PMs@NaF-SAP was obtained by mixing 3-maleimidopropionic acid-poly(ethylene glycol)-block-poly(L-lysine)/phyenylboronic acid (MAL-PEG-b-PLL/PBA) with tannic acid in NaF aqueous solution and conjugating salivary peptide (SAP). This (Figure 6c) was co-loaded with an antibacterial agent and restorative agent. Low pH could crack the borate ester bond, so that SAP could enable the nanoparticles with a strong adhesion to the enamel. The PMs@NaF-SAP was attached strongly to the tooth surface and intelligently released drugs at the acidic pH, thus protecting the tooth surface from the cariogenic biofilm (Figure 6d) and restoring the microstructure of the demineralized site at the same time [63].

Drug delivery systems that can simultaneously respond to multiple environmental stimuli have also been developed [7]. Peptide pHly-1 nanoparticles were sensitive to both pH and lipids [65]. Under neutral physiological conditions, pHly-1 maintained a β-folded conformation. However, in an acidic cariogenic microenvironment, pHly-1 underwent a helical conformation transition by binding with the bacterial membrane, which could destroy the bacterial cell membrane and kill cariogenic bacteria.

Moreover, silica-based mesoporous bioactive glass, with pore sizes of 2 to 50 nm, can also serve as a carrier for a drug delivery system. These glass particles are also known as mesoporous silica nanoparticles (MSNs) [47]. MSNs have several advantages, including high surface area, high pore volume, unique pore structure characteristics, and uniform pore size distribution. MSNs loaded with CHX were grafted with polyglycolic acid (PGA) (Figure 6e) and then added to a commercial adhesive, which maintained excellent bond strength [54]. The CHX release from the nanoparticles was pH sensitive. These nanoparticles could penetrate dentin tubules to release CHX inside dentin (Figure 6f) and play their role of intelligent antibacterial effect. In another study, MSNs were assembled in adhesives containing several broad-spectrum antibacterial agents to accommodate as many payloads as possible while achieving stable and long-term release [66]. MSNs could effectively stabilize antibacterial agents in dental resins. When affected by biodegradation, MSNs are released in response to saliva and bacterial degradation factors, making the material responsive to cariogenic bacteria. Further efforts are needed to combat cariogenic bacteria by modifying the drugs loaded by MSNs as well as improving the microstructure of MSNs.

## 7. Conclusions

This article represents the first review on novel smart pH-responsive dental resin materials to combat caries, and on the cutting-edge development of autonomously responsive smart dental materials (Listed in Table 1). These materials intelligently respond to changes in the local oral pH caused by cariogenic bacteria. They can buffer pH changes by releasing therapeutic ions, increasing the release at low pH when these ions are most needed, promote local remineralization, inhibit or kill bacteria and regulate the oral microbial community to resist caries, and trigger the antibacterial activity only at low pH. Several new smart materials were incorporated in dental composites as active ingredients, as modified glass fillers and as modified resin monomers. In addition, the use of poly(amido amine) in conjunction with NACP significantly enhanced the acid neutralization and remineralization of NACP-containing dental resins. Furthermore, the development of smart materials for treating periodontal inflammation based on the enzyme response system also showed great promise. Further research is needed to improve the smart dental resins to be able to respond to a variety of physical, chemical, and biological changes in the local environment to prevent secondary caries and inhibit acidogenic bacteria. Additionally, further effort is also needed to develop intelligent dental materials that are biocompatible and locally precise in their ability to repair damaged dental tissues and reverse localized dental caries.

## Figures and Tables

**Figure 1 polymers-15-02611-f001:**
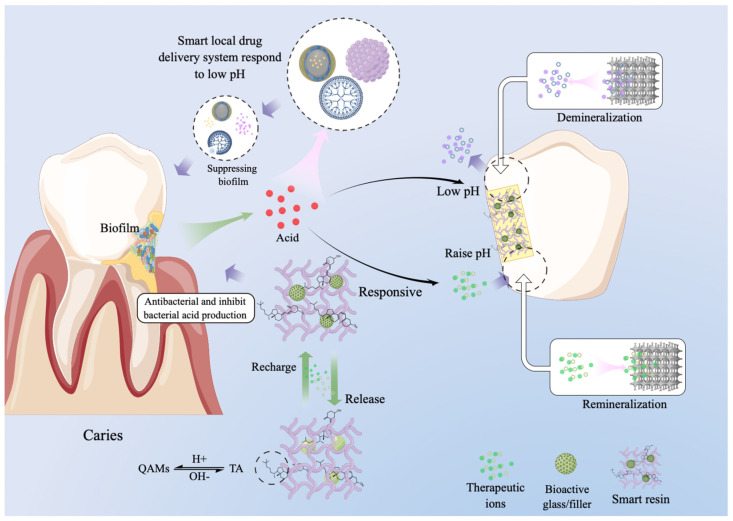
Schematic for dental smart materials. Bacteria adhere to tooth hard tissues to form plaque biofilms. Biofilms metabolize sugars and produce acids, reducing local pH and demineralizing tooth tissues. Smart dental materials respond to low pH and release therapeutic ions, which will raise the pH and promote remineralization. Furthermore, these smart materials can inhibit biofilm growth and diminish acid production. Therapeutic ions can be recharged and re-released to achieve a sustained and long-term remineralizing effect. In addition, smart drug delivery systems can respond to acids produced from bacterial metabolism by releasing loaded antibacterial agents. (Draw by Figdraw).

**Figure 2 polymers-15-02611-f002:**
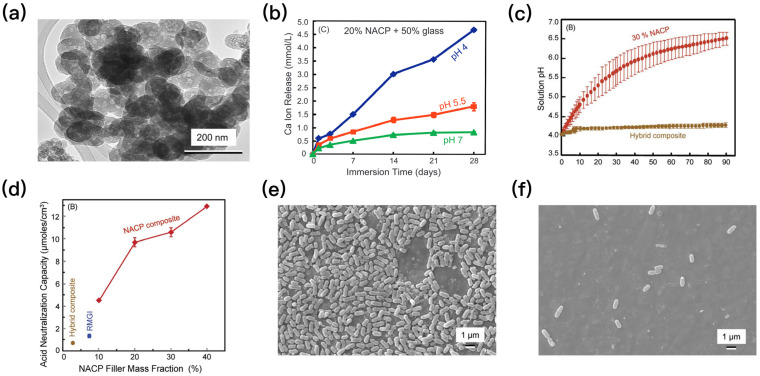
(**a**) A typical TEM micrograph of the NACP particles. (**b**) Ca ion release from the nanocomposite filled with 20% NACP and 50% glass. Ca ion release increased with decreasing the solution pH. (**c**) NACP raised the pH of a cariogenic acid solution (pH 4). The nanocomposite with 30% NACP greatly increased the pH. The commercial composite has little increase in pH. (**d**) NACP had acid neutralization capacity. It was evaluated by calculating the amount of a base (potassium hydroxide) that would need to be added to the pH 4 solution in order to increase the pH. (**e**) SEM micrographs of *S. mutans* on hybrid commercial composite and (**f**) nanocomposite with 40% NACP. The hybrid commercial composite was nearly entirely covered by bacteria. The nanocomposite had the least bacteria. (Adapted from ref. [17,20], with permission).

**Figure 3 polymers-15-02611-f003:**
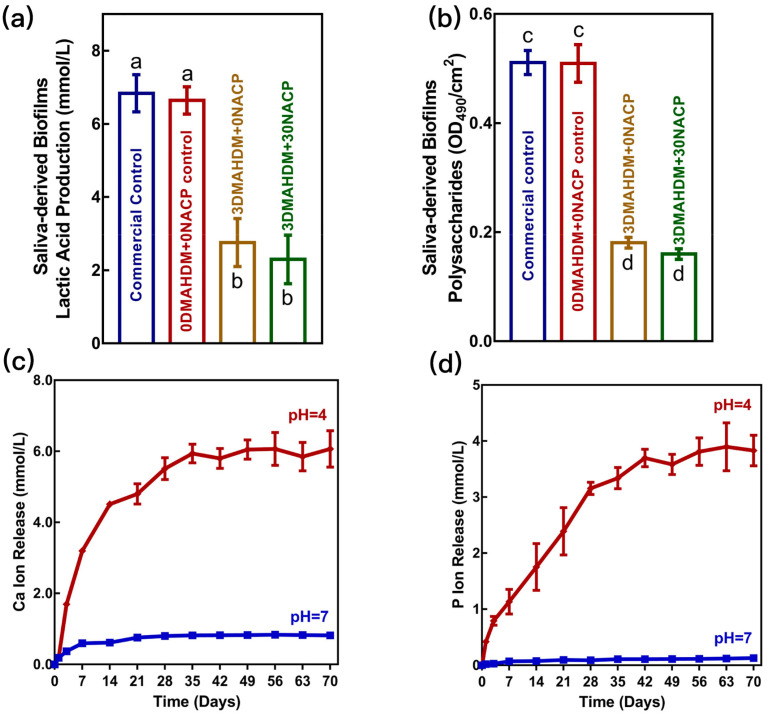
Inhibition effects of composites against the cariogenic activities of saliva-derived biofilms. (**a**,**b**) Lactic acid and polysaccharide production by saliva-derived biofilms. The acid and polysaccharide production was significantly reduced by the composites with 3DMAHDM + 0NACP and 3DMAHDM + 30NACP. (**c**,**d**) Calcium (Ca) and phosphate (P) ion releases from the 3% DMAHDM + 30% NACP composite immersed in solutions of pH 4 and 7. The ion release substantially increased from pH 4 to 7. (Adapted from ref. [32], with permission.).

**Figure 4 polymers-15-02611-f004:**
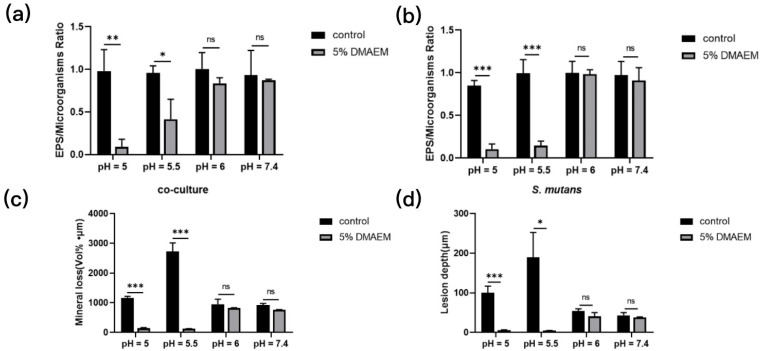
DMAEM@RA decreased the production of EPS of the *S. mutans* and dual-species biofilms at pH 5 and 5.5. (**a**,**b**) Quantitative analysis of the ratios between the EPS and the microorganisms in the 5% DMAEM@RA groups and the control groups at different pH values. DMAEM@RA inhibited the tooth demineralization caused by *S. mutans* and *C. albicans* dual-species biofilms in a pH-responsive way in vitro (* *p* < 0.05, ** *p* < 0.01, and *** *p* < 0.001; ns, no statistical significance). (**c**,**d**) The values of mineral loss and lesion depth of different groups after treatment for 3 days (* *p* < 0.05 and *** *p* < 0.001; ns, no statistical significance). (Adapted from ref. [43], with permission).

**Figure 5 polymers-15-02611-f005:**
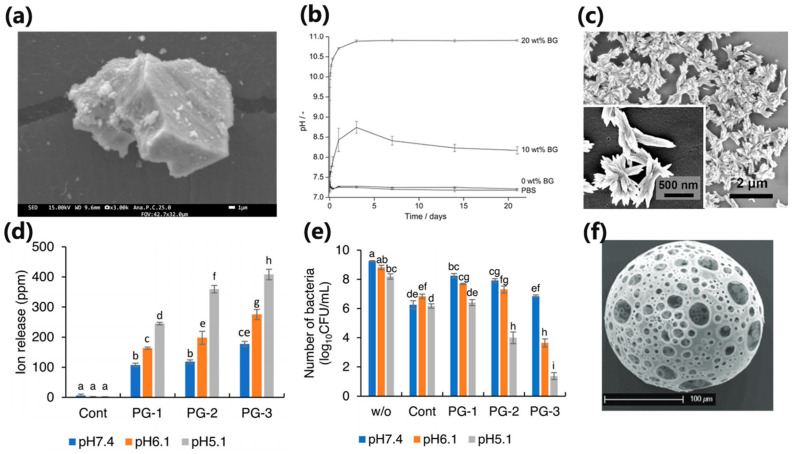
(**a**) FE-SEM image of the bioglass particle surface. (**b**) The cured specimens containing 10 wt.% BG induced a pH increase to 8.7 in PBS, which then decreased slightly. In contrast, 20 wt.% BG loading resulted in a pH of 10.8 in PBS, which stayed constant during the 21-day period. (**c**) SEM images of 20 wt.% loading of BG in Bis-GMA/TEGDMA after immersion in PBS for 21 days, showing calcium phosphate precipitates. (**d**) With a decrease in pH values, the release of Zn^2+^ increased significantly by Zn^2+^-releasing glass in the pH-adjusted BHI broth (*p* < 0.05) (PG-1, PG-2, PG-3 represented the Zn^2+^ contents, and the larger the number, the higher the Zn^2+^ content). (**e**) Number of viable *S. mutans* after 24 h incubation at pH 5.1, 6.1, and 7.4 with different content of glass particles (*p* < 0.05) (PG-1, PG-2, PG-3 represented the Zn^2+^ contents, and the larger the number, the higher the Zn^2+^ content). (**f**) Representative SEM images of PBGSs obtained by flame spheroidization. Glass compositions of P40 (40P_2_O_5_–16CaO–20Na_2_O–24MgO, in mol %). (Adapted from refs. [47,48,49], with permission.).

**Figure 6 polymers-15-02611-f006:**
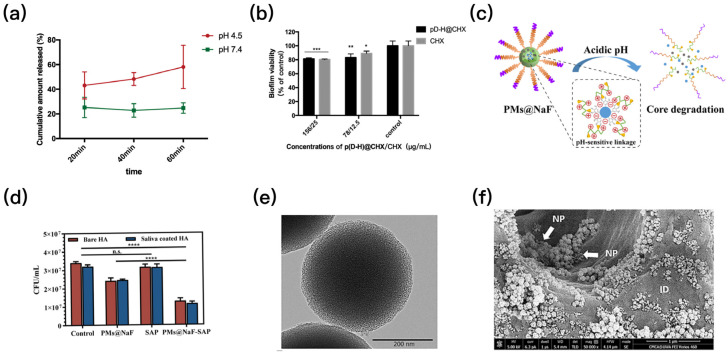
(**a**) The p(DH)@CHX can release CHX in response to pH. The release of CHX in acidic environments was significantly higher than that in neutral environments. (**b**) The p(DH)@CHX showed antibacterial effects on *S. mutans* biofilms, and the biofilm metabolic activity was measured using the MTT assay (* *p* < 0.05, ** *p* < 0.01, *** *p* < 0.001). (**c**) Proposed concept of pH-responsive biomaterial PMs@NaF. (**d**) The PMs@NaF-SAP reduced bacterial biomass, indicating the potent ability of providing anti-bacterial adhesion and cariogenic biofilm resistance (**** *p* < 0.0001). (**e**) TEM image for CHX-loaded/MSN. (**f**) SEM image. The CHX-loaded/MSN has the potential to penetrate dentin tubules to release CHX, NP: nanoparticles, ID: intertubular dentine. (Adapted from refs. [53,57,63], with permission).

**Table 1 polymers-15-02611-t001:** Type, component, and features of smart dental resin.

Type	Component	Features
Composites	32% BT(BisGMA+TEGDMA) + 35% Glass particles + 3% DMAHDM+30% NACP [33,34]	Inhibit bacteria; inhibit enamel demineralization; increase Ca and P ion release at low pH
	35-38% UV (UDMA+TEG-DVBE) + 2%–5% DMAHDM + 20% NACP + 43% glass [36,37,38]	Inhibit bacteria; remineralize; lower shrinkage-stress
	25% (BisGMA+TEGDMA) + 75% TTCP [31]	Increase Ca and P ion release at low pH
Adhesive	PEHB&PM primer, containing 5% MPC + 5% DMAHDM + 30% NACP [17]	Reduce protein adsorption and bacterial adhesion; inhibit bacteria; increase Ca and P ion release at low pH
	SBMP adhesive containing 5% DMADDM + 0.1% NAg + 20% NACP; SBMP primer containing 5% DMADDM + 0.1% NAg [30]	Inhibit bacteria; increase Ca and P ion release at low pH
	Clearfil SE Bond containing 5% TAs (DMAEM, HMAEM) [43,44]	Inhibit bacteria; increase Ca and P ion release at low pH
	Scotchbond™ bond (3M ESPE) + 5% CHX-loaded/MSN-PGA [54]	Inhibit bacteria; pH-response
	Adper Scotchbond Multi-Purpose Adhesive (SBMP) + 10% OCT-DMSNs (octenidine dihydrochloride, OCT) [66]	Inhibit bacteria; pH-response
Resin infiltrant	BisGMA + TEGDMA + 5% DMAEM [46]	Inhibit bacteria; pH-response
Resin sealant	ClinproTM Sealant (3 MTM ESPETM) + 2.5–10% DMAEM [45]	Inhibit bacteria; pH-response; reduce microleakage

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
