# Peer review of "Smart Dental Materials Intelligently Responding to Oral pH to Combat Caries: A Literature Review"

_polymers, 2023, doi:10.3390/polym15122611_

Round 1

Reviewer 1 Report

In the title, after it, it should be added that it is a revision and of what type.

The text has several grammatical errors in English. I pointed out some but the authors must undergo a rigorous review.

Abstract:  two final points after “ ….tooth structures..!

L38: In plural:  “Efforts have been made to improve tooth cavity filling materials, including glass ionomer cements (GIC), resin-modified glass ionomer cement(RMGIC) and composites. “

- Despite being a simple review, the authors should minimally indicate what were the criteria for searching and selecting articles; not being a systematic review or just a scoop review, the research must be minimally systematized (years of research, keywords or mesh terms, type of articles, language...)

- Refer and discuss to whether these materials also act and have some effectiveness in the face of more generalized and aggressive stimuli such as very erosive acid drinks or gastro esophageal reflux, for example.

- L43:  Recently, smart dental materials have been developed that can function in response to specific changes occurring in the oral environ- 44 ment, such as light, pH, and temperature [4-6]. Several dental resins were recently re- 45 ported to be responsive to local pH in order to combat caries (Fig. 1).

But the launch of this type of material is not recent, it has been many years, more than 20, that there were other materials that did not work well; What has changed now? see for example Ariston pHc from Ivoclar Vivadent

- The exchange of ionic components does not imply a lower structural stability of the material, impairing some of its mechanical properties and consequent clinical performance (such as resistance to fracture, wear or even aesthetic properties). Discuss this topic please.

L57 To date, 57 a literature search revealed no review paper on the smart pH-responsive dental resin ma- 58 terials. The present study represents the first review article on the development and mech- 59 anisms of smart pH-responsive dental resin materials. 

This is not entirely true! See Maloo 2022; McCabe 2011, Francois 2020. These articles are very correlated with this theme and are not even included in the bibliography.

L72:  “....decline in plaque pH and…” Not”…palque pH…”

L72: “….sustained low pH environment often cause …..” Not “….envieonment….”

L120.  “…pH (more detailed explanation in Chapter 5).”   Chapter 5?  An article is not organized by chapters. It should be “...at point 5”

Fig. 2 : there is a lack of supporting references and the origin of graphic schemes and images.

L143:  bibliographical references for the statements contained in line 140 to 153

L169: “There are more evidences provided by several other studies, which implicated the wide applicability in other biomaterials. Novel bioactive crown cement and magnetic nanoparticle-containing adhesive shows similar antibacterial and remineralizing capabilities.”  

Several studies?! Where are the references for these multiple studies?

Fig. 3:  there is a lack of supporting references and the origin of graphic schemes and images.

Fig. 4: there is a lack of supporting references and the origin of graphic schemes and images.

L229:  “...bond with the host boneto repair bone” . correct for “…bond with the host bone to repair bone”

Fig. 5: there is a lack of supporting references and the origin of graphic schemes and images.

L297:  “...controlled and locolized release of drugs….” 

Fig. 6: there is a lack of supporting references and the origin of graphic schemes and images.

L360: “This article represents the first review on novel smart pH-responsive dental resin materials to combat caries”.  For sure? Please see previous comments.

References 

There are too many self-citations.

There are many grammatical errors in English (I pointed out a few)

Author Response

Dear Skyler,

I hope you are doing very well.  Thank you very much for a favorable review on our paper, polymers-2383092, entitled: “Smart dental materials intelligently responding to oral pH to combat caries: a literature review”. The reviewers provided excellent comments, all of which have been addressed in the revised paper.  Below is a point-by-point response to the review comments.  Each comment is followed by our response in a red color. The revised sections in the paper are also highlighted in red.Please see the attachment

We look forward to hearing from you.  Thank you for your help.

Best Regards,

Ke Zhang

Professor, Department of Orthodontics, School of Stomatology

Capital Medical University

Reviewer 2 Report

This manuscript reviews the role of biomaterials which can respond to and reverse acidic environments in the oral cavity to prevent the degradation of dental tissue. I enjoyed reading this review and commend the effort to contribute to this important and evolving field. The content expands on the existing literature but is not the first to review the topic as stated in the manuscript. I would recommend publication if the following points were addressed: 

  1. The assertion made in the introduction that there is no existing review on pH-responsive dental materials needs to be revised. The topic was covered in a 2018 paper (https://doi.org/10.1038/s41413-018-0032-9). I suggest revising this statement to correctly situate your work within the current literature, possibly indicating how your paper expands or diverges from the previous work.
  2. References in figures: It would be beneficial to provide references within your figure legends if the images or data have been taken from other sources. 
  3. Table references: Including references in the table will enhance the reader's ability to locate the original papers for further reading.
  4. Clarity of Figure 1: The current presentation of Figure 1 is difficult to follow due to the excessive use of arrows in multiple directions. I recommend redesigning this figure to create a more logical and straightforward flow that the reader can easily comprehend.

Author Response

Dear Skyler,

I hope you are doing very well.  Thank you very much for a favorable review on our paper, polymers-2383092, entitled: “Smart dental materials intelligently responding to oral pH to combat caries: a literature review”. The reviewers provided excellent comments, all of which have been addressed in the revised paper.  Below is a point-by-point response to the review comments.  Each comment is followed by our response in a red color.  The revised sections in the paper are also highlighted in red. Please see the attachment.

We look forward to hearing from you.  Thank you for your help.

Best Regards,

Ke Zhang

Professor, Department of Orthodontics, School of Stomatology

Capital Medical University

Round 2

Reviewer 1 Report

It seems to me that the authors have solved the issues and corrections requested in a very satisfactory way. However, I have just one last concern: with the alteration of the bibliography and, consequently, the numbering of each citation, alterations were made to the citations throughout the article that seem to have been done "manually" and not automatically electronically with software such as EndNote or Mendeley ! Was it the case? If yes, they should do a last detailed review of all citations in the text so that no errors remain. But ideally they should do it electronically.